# Trend and determinants of quality of family planning counseling in Ethiopia: Evidence from repeated PMA cross-sectional surveys, (2014–2019)

**Bedilu Alamirie Ejigu**[1]*, **Assefa Seme**[2], **Linnea Zimmerman**[3], **Solomon Shiferaw**[2]

**1** Department of Statistics, Addis Ababa University, Addis Ababa, Ethiopia, **2** School of Public Health, Addis Ababa University, Addis Ababa, Ethiopia, **3** Department of Population, Family and Reproductive Health, Johns Hopkins Bloomberg School of Public Health, Baltimore, MD, United States of America

* bedilu.alamirie@aau.edu.et

**Data Availability Statement:** All data used in this publication are publicly available at https://www.pmadata.org/data/request-access-datasets.

## Abstract

### Introduction

The modern contraceptive prevalence rate (mCPR) among married women has increased by nearly five-fold in Ethiopia from 8.1% in 2000 to 37% in 2019. Despite this increase, receipt of high quality contraceptive counselling, as measured by the percentage of contraceptive users who were told about other methods, counseled on side effects and counseled what to do in the event that they encountered side effects, has declined in recent years. The quality of family planning counseling service measured by using these three components, known as the Method Information Index(MII), is an index designed to measure quality and a key indicator of the FP2020 initiative. The effects of potential client and service provider-level factors on receipt of high quality counseling and its progress over time have not been well studied in Ethiopia.

### Methods

We pooled data from seven Performance Monitoring for Action (PMA), formerly PMA2020, survey-rounds to examine the trend and effect of potential factors on receiving high quality of family planning counseling service in Ethiopia. Data from a total of 15,597 women aged 15 to 49 from seven survey-rounds were used in the analysis. To account for the study design and unequal probabilities of selection from target-populations for sampled women, design-based analysis was used to compute proportions. Multilevel ordinal regression model with enumeration area as a second level were employed to examine potential factors associated with quality of family planning counseling service.

### Results

We found that the percentage of women who received high quality family planning counseling service declined from 39% (95%CI: 33%, 44%) in 2015 to 12% in 2019 (95% CI: 10%, 14%) nationally. Amhara region had the lowest percentage of women receiving high quality

Citations for the data are Addis Ababa University, School of Public Health at the College of Health Sciences and The Bill & Melinda Gates Institute for Population and Reproductive Health at The Johns Hopkins Bloomberg School of Public Health.

**Funding:** This work was supported, in whole, by the Bill and Melinda Gates Foundation [INV 009466]. Under the grant conditions of the Foundation, a Creative Commons Attribution 4.0 Generic License has already been assigned to the Author Accepted Manuscript version that might arise from this submission.

**Competing interests:** The authors have declared that no competing interests exist.

counseling at both the earliest(2014) and latest(2019) survey rounds(17% and 6%, respectively). Results show that lack of media exposure about family planning, having no formal education, using short-acting methods, and getting the service from pharmacy were the main factors associated with receiving low quality family planning counseling service.

## Conclusions

Given the importance of continuous provision of information on the range of family planning methods, it is imperative to use media and particularly regional media which can effectively address the rural populations in local languages as an important vehicle of information on family planning. Interventions aimed at improving quality of family planning counseling need to be mindful of regional disparities in the severity of the problem to ensure equity in service access. To improve the coverage of high quality family planning counseling service, there is an urgent need to re-visit the format of family planning counselling services.

## 1 Background

The 2030 Sustainable Development Goals (SDGs) calls on countries "by 2030, to ensure universal access to sexual and reproductive health-care services, including for family planning, information and education, and the integration of reproductive health into national strategies and programmes" [1]. In 2019, among 1.9 billion women of reproductive age (15–49 years) worldwide, 1.1 billion had a need for family planning [2]. Of these 842 million were users of modern contraceptive methods and 190 million women who wanted to avoid pregnancy were not using any contraceptive method [1]. In sub-Saharan Africa, annual rate of changes (percentage points) in modern contraceptive use among women of reproductive age varied substantially from 0.77 to 3.17 [3].

Counseling plays a key role in enhancing family planning services, providing contraception related information and supporting family planning and fertility goals for women of childbearing age [4]. The Method Information Index (MII) is the current standard proposed by the Track 20 initiative to monitor quality of family planning counseling [5, 6]. The simple index uses three questions—whether the respondent received counseling on multiple contraceptive methods, whether she was told about side-effects, and if so, whether she was told what to do about side-effects—that are available in the majority of population-based surveys, including the Demographic and Health Survey (the DHS) and Performance Monitoring for Action (PMA) [7]. Adaptations have been proposed to the MII, including a fourth question assessing whether women were told they could switch to another method. This question was added in PMA 2019 Survey, however, and for trend purposes is not included in the current analysis.

Despite substantial global efforts to improve conceptualization, measurement, and delivery of high-quality family planning services, studies consistently find that receipt of high-quality services is low. In a study exploring the levels and trends of the MII across 25 countries, urban residence, education, household wealth, and method type were generally found to be associated with receiving higher quality care [8]. In all countries combined, the median MII values show that the contraceptive information received by women varied by method type and was highest for implant users and lowest among women relying on sterilization.

In Ethiopia specifically, despite increases in contraceptive coverage over time the percentage of contraceptive users who received all components of Method Information Index(MII)

has not changed substantially [9, 10]. Studies on family planning counseling services in Ethiopia consistently show that high quality of family planning counseling service coverage is below 40% [11–14].

Understanding which individual, social, and health system factors are associated with the receipt of high-quality family planning counseling is critical in order to identify inequities and improve overall service delivery. Multiple studies have assessed facility and provider characteristics that are associated with the receipt of quality family planning counseling services [15–18], but relatively few studies have assessed the client-level and social factors that may be related to receipt of high quality counseling service.

A previous study in Ethiopia found that in addition to these factors, women who received family planning services from a public provider had higher odds of receiving all components of the MII. Little additional evidence exists, however, on the client-level factors that may be associated with receiving high quality services, resulting in a call for additional research into identifying factors associated with quality [19].

Studies that have explored client-level and social factors associated with higher quality services have generally been limited by use of one to two cross-sectional surveys which limit the ability to detect trends in receipt of high quality family planning counseling service over time. Increasingly, however, it is common to see repeated cross-sectional surveys carried out in multiple rounds over short periods of time, particularly surveys conducted by the Performance Monitoring for Action (PMA) project. Pooling such data improves the power of test due to an increased sample size and can be used to better investigate trends over-time. A study done by [12] used pooled data from five survey rounds in Ethiopia to compute the percentage of women who received high counseling service across survey years by different factors. However, the analysis on factors of quality of family planning counseling service was limited only to one survey round (i.e., 2018 survey). To our knowledge, there is no study that considered repeated cross-sectional surveys to identify factors related to quality of counseling service in Ethiopia.

In the present study, we use repeated cross-sectional data from seven-rounds of a national survey to: i) establish the trends in quality of family planning counseling among all women aged 15–49 years, ii) identify client-level and social factors associated with quality of counseling, and iii) identify regional disparities in quality of counseling. Findings from this study can help to inform planning interventions to improve the quality of family planning counseling services in Ethiopia.

## 2 Materials and methods

### 2.1 Study design and data source

The data for this study were obtained from seven rounds of the Performance Monitoring for Action (PMA) surveys, which were undertaken yearly since 2014 (formerly as Performance Monitoring and Accountability 2020 (PMA2020)). The PMA surveys are repeated cross-sectional surveys based on a multistage stratified cluster sampling design with urban-rural stratification for Amhara, Oromia, Tigray, and Southern Nations, Nationalities, and Peoples' (SNNP) regions, and for the remaining regions, regions served as the strata, without additional urban/rural stratification. The primary sampling units or Enumeration Areas (EAs) are selected using probability proportional-to-size method for which the sample selection probability depends on the size of population. For the first round (referred to as PMA2014/Ethiopia) and latest round considered in this study (referred to as PMA2019/Ethiopia) of data collection the survey conducted in 200 and 265 enumeration areas, respectively, which were selected by the Central Statistics Agency (CSA) to be representative at the national and regional level. Prior to data collection, all households in each enumeration area were listed and mapped by the resident

enumerators to create a frame for the second stage of the sampling process. This mapping and listing process was done in the first week of data collection in each enumeration area. Once listing was completed, 35 households were randomly selected by field supervisors using a phone-based random number-generating application. All occupants in selected households were enumerated and from this list, all eligible women between the ages of 15 to 49 were administered the female-household questionnaire after informed consent was obtained. In PMA surveys considered in this analysis, the response rate is high across different survey years which ranges from 95.6% to 99.2%. In all survey-rounds, data were collected using smartphones.

The details of the PMA2020 survey design and sampling techniques are described elsewhere [20], while updates made to the sampling design in 2019 are further described by Zimmerman and colleagues [21]. PMA uses standard data collection procedures which are consistent across-rounds, and consistent in content, allowing pooling the data and making comparisons over time. To assess trend over time and identify factors linked with the outcome variable, described below the pooled data from seven-rounds, spanning 2014 to 2019 were considered.

## 2.2 Sample

In this study, data from nationally representative cross-sectional female and household surveys collected on yearly bases from 2014 to 2019 among women aged 15–49 were utilized. A total of 15,597 women who were asked about the three questions, described below, were considered as our analytical sample.

## 2.3 Variables

The outcome variable in this study, quality of counseling service, was derived based on the responses to the three equally weighted questions used in the MII. Women were asked to report whether, at the visit where they first received their current contraceptive method, they were: i) told about other methods, ii) counseled on side effects, and iii) counseled on what to do if she experienced side effects. Originally, the data related to the variable of interest were collected using possible Yes or No questions to assess quality of counseling. Using these questions, the MII was assessed two separate ways; first, as a binary variable, indicating whether a woman received all components of the MII ("high-quality") or not, and second, as an ordinal variable. We defined the ordinal variable with four categories indicating "high" quality (all three components received), "moderate" quality (any two of three), "low" quality (any one of three), and "no" counseling (no components received).

In order to assess the effect of different factors on receiving high quality family planning counseling service, the outcome variable for the $i^{th}$ women from $j^{th}$ enumeration area(EA), $Y_{ij}$, is dichotomized as follows:

$$y_{ij} = \begin{cases} 1 & \textit{if a woman got all the three counseling services (high counseling)} \\ 0 & \textit{if a woman didn't get the three counseling service}. \end{cases}$$

This categorization was used to compute design-based prevalence of high quality family planning counseling service by different socio-demographic factors, described further below.

The analysis using the dichotomized outcome cannot provide information about different counseling levels simultaneously, thus further disaggregation was done based on the following

**Table 1. Description of exploratory variables considered in the study.**

| Variable | Variable description and/or categorization |
|---|---|
| Age | Classified as: 15–19 20–24, 25–34, 35–49 |
| Education | Never attended, Primary, Secondary/higher |
| Parity | Classified as: 0 children, 1–2 children, 3+ children |
| Method type | Categorized as: long-acting and short-acting |
| Wealth | Household wealth quintile (lowest to highest) |
| Media exposure[†] | Exposure to family planning information disseminated through media |
| Method source | Source of method (hospital, health post, health center, pharmacy, other) |
| Residence | Place of residence: urban, rural |
| Region | Tigray, Amhara, Oromia, SNNP, Addis Ababa, Other[‡] |
| Survey year[⁋] | Survey year considered as categorical variable |

[†] Composite variable constructed using YES or NO question: have you heard/seen or received message about family planning from radio, TV, newspaper or magazine, social media, text message on mobile.

[‡] Other regions represents the six small regions, namely Afar, Somali, B-Gumuz, Gambella, Harari and Dire Dawa.

[⁋] Round 1 and 2 surveys were done in 2014.

categorization:

$$y_{ij} = \begin{cases} \text{No counseling} & \text{if a woman didn}\prime\text{t get any service} \\ \text{Low counseling} & \text{if a woman got one of the services} \\ \text{Medium counseling} & \text{if a woman got two of the services} \\ \text{High counseling} & \text{if a woman got three of the services} \end{cases} \quad (1)$$

After reviewing related literature, possible factors linked with the quality of family planning counseling were extracted from the survey data. Table 1 presents the list of factors we hypothesize are associated with the receipt of quality of family planning counseling service considered in this study. We define long-acting methods as female sterilization, implant and IUD and short-acting methods as the injectable, pill, emergency contraception, male condom, the Standard Days Method, and lactational amehorrhea method (LAM).

## 2.4 Statistical analysis

Nationally representative survey data-sets are often complex in nature for two reasons: i) the use of stratified multistage cluster sampling, and ii) unequal probabilities of selection from target-populations for sampled elements, often as a result of oversampling of key subgroups. Thus to account for the study design and unequal probabilities of selection from target-populations for sampled women, sampling weights were used during the estimation of proportions and their respective confidence interval [22].

The types of models used for data analysis depend on the nature and measurement scale of the outcome variable under consideration. The level of measurement of the variable is both binary and ordinal. When the response categories are ordered, the use of this ordering yields more parsimoniously parameterized models [23]. In such models, the ordinal nature of the response is taken into account by considering the cumulative probabilities. The data from different women are assumed to be independent, but due to the clustered nature of the data, observations from the same cluster/EA may not be independent.

Women living in the same EA may share similar health facilities, and the quality of counseling services available to them may be more alike than women living in another EA. Hence, our

modeling approach should takes into account the correlated nature of the data by introducing a random effects term to account for a variety of situations, including EA heterogeneity, and unobserved covariates within that EA. As a result, in this study we employ a random effects model for ordinal outcomes.

Let $\mu_{ij}$ be the probability of the $i^{th}$ subject from the $j^{th}$ EA being in the response category $k$, $\mu_{ij} = P(y_{ij} = k)$. Further, we let cumulative probability of the response in category $k$ or above represented by $\Pi_{ijk} = P(y_{ij} \geq k)$. The random intercept proportional odds model (POM) is given by:

$$logit(\Pi_{ijk}) = log\left(\frac{P(Y_{ij} \geq k|X'_{ij})}{P(Y_{ij} < k|X'_{ij})}\right) = \beta_{0k} + X'_{ij}\beta + b_j \qquad (2)$$

where $k$ is the level of the ordered category. The parameter $\beta_{0k}$ is the intercept for category $k$, usually is considered as nuisance parameter of little interest ([23]), $X_{ij}$ is a vector of exploratory variables described in Table 1 with associated regression coefficients $\beta$. The random intercept $b_j$ is assumed to vary randomly among EA's according to a $N(0, \sigma_2^2)$ distribution. The variance of the random effect ($\sigma_2^2$) represents how much variability there is between clusters with respect to the response variable.

Model 2 assumes an identical effect of the predictors for each cumulative probability. The model assumes that, conditional on the random effects, the coefficients that describe the relationship between, for example, the lowest versus all higher categories of the response variable are the same as those that describe the relationship between the next lowest category and all higher categories, etc. This is called the proportional odds assumption or the parallel regression assumption [23]. Specifically, the model implies that odds ratios for describing effects of explanatory variables on the response variable are the same for each of the possible ways of collapsing the response to a binary variable. The proportional odds (PO) assumption for each variable were tested using the *Brant* test ([24]), and held across all variables.

Further, to quantify the relative importance of clusters as a source of variation about receiving high counseling service, EA-level random effects variance was expressed in terms of Variance Partition Coefficients (VPC).

The proportion of total variation of receiving high family planning counseling attributable to EA random effect ($VPC_{\sigma_2^2}$) is computed as follows [25, 26]:

$$VPC_{\sigma_2^2} = \frac{\sigma_2^2}{\sigma_2^2 + \frac{\pi^2}{3}},$$

where $\sigma_2^2$ is EA (level-2) random effect variance, and $\pi^2/3 = 3.29$ is individual level variance is equal to the variance of a logistic distribution [27]. In Model 2, fixed effects parameter has a conditional interpretation. It refers to the consequence of changing the value of an explanatory variable, for which the fixed effect is the coefficient, for a given value of the random effect and the other fixed effects.

In all exploratory data analyses, sample weights calculated based on the multistage sampling design were considered. Data analyses were done using Stata 16.1 software [28], and graphs presented in this manuscript were generated using the **R** software [29] by *ggplot*2 package [30]. In this study, for statistical significance, the considered alpha(type-I error) is 0.05.

## 2.5 Ethics approval statement

The study protocol was approved by Institutional Review Boards of College of Health Science at Addis Ababa University, Ethiopia and at the Bloomberg School of Public Health at Johns Hopkins University in Baltimore, USA.

# 3 Results

## 3.1 Descriptive results

Table 2 presents client, service provider, and regional level characteristics of the sampled study population by year. Across all years, the majority of women lived in rural areas. Approximately 50% of respondents were age 25–34 years, and less than 20% of respondents were nulliparous. On average, around 50% of women got received their contraceptive methods from health centers. The use of long-acting contraception methods (female sterilization, implant and IUD) increased from 21.81% in 2014 to 36.94% in 2019. The percentage of respondents who

**Table 2. Sample characteristics of PMA survey respondents across survey years.**

| Factor | | Survey Round | | | | | |
|---|---|---|---|---|---|---|---|
| | | 2014 † | 2015 | 2016 | 2017 | 2018 | 2019 |
| Total (n) | | 3889 | 2450 | 2503 | 2366 | 2353 | 2036 |
| Residence | Urban | 40.11 | 40.86 | 42.67 | 42.39 | 43.43 | 45.38 |
| | rural | 59.89 | 59.14 | 57.33 | 57.61 | 56.57 | 54.62 |
| Region | Tigray | 17.82 | 14.94 | 13.62 | 14.45 | 14.28 | 11.59 |
| | Amhara | 24.79 | 20.98 | 22.97 | 21.89 | 22.95 | 22.20 |
| | Oromia | 12.81 | 19.71 | 19.50 | 19.32 | 20.82 | 22.20 |
| | SNNP | 26.10 | 26.12 | 26.01 | 24.43 | 24.14 | 20.43 |
| | Addis Ababa | 12.45 | 12.12 | 11.91 | 11.88 | 11.64 | 9.72 |
| | Other | 6.04 | 6.12 | 5.99 | 8.03 | 6.16 | 13.85 |
| Education | Never attended | 32.57 | 30.73 | 30.78 | 29.24 | 28.07 | 32.12 |
| | Primary | 37.85 | 37.19 | 37.73 | 36.48 | 36.80 | 39.15 |
| | Secondary/higher | 29.58 | 32.08 | 31.49 | 34.28 | 35.14 | 28.73 |
| Age(year) | 15–19 | 7.02 | 7.35 | 7.03 | 8.62 | 6.84 | 7.66 |
| | 20–24 | 22.68 | 24.86 | 22.41 | 22.36 | 20.95 | 20.97 |
| | 25–34 | 47.06 | 45.10 | 46.66 | 45.73 | 47.22 | 46.66 |
| | 35–49 | 23.25 | 22.69 | 23.89 | 23.29 | 24.99 | 24.71 |
| Parity | 0 Children | 15.66 | 18.82 | 15.26 | 17.97 | 17.00 | 10.76 |
| | 1–2 children | 42.76 | 41.67 | 44.19 | 43.38 | 42.07 | 46.39 |
| | 3+ children | 41.58 | 39.51 | 40.55 | 38.65 | 40.93 | 42.85 |
| Method source | Hospital | 17.26 | 17.51 | 16.42 | 20.63 | 21.76 | 9.04 |
| | Health post | 19.64 | 19.28 | 17.80 | 20.45 | 18.53 | 22.79 |
| | Health center | 50.42 | 48.86 | 52.4 | 44.96 | 43.56 | 60.17 |
| | Pharmacy | 6.62 | 8.59 | 7.96 | 9.19 | 11.39 | 6.43 |
| | Others | 6.06 | 5.77 | 5.42 | 4.77 | 4.76 | 1.57 |
| Method type | Short-acting | 78.19 | 75.21 | 73.79 | 72.03 | 69.62 | 63.06 |
| | Long-acting | 21.81 | 24.79 | 26.21 | 27.97 | 30.38 | 36.94 |
| Media exposure | No | 37.80 | 36.32 | 38.50 | 44.14 | 44.26 | 53.93 |
| | Yes | 62.20 | 63.68 | 61.50 | 55.86 | 55.74 | 46.07 |
| Wealth quintile | Lowest | 7.02 | 8.9 | 8.99 | 9.55 | 9.82 | 11.89 |
| | Lower | 9.39 | 9.22 | 10.31 | 9.86 | 10.84 | 15.28 |
| | Middle | 11.03 | 10.86 | 12.78 | 10.23 | 10.62 | 17.83 |
| | High | 17.56 | 17.76 | 16.74 | 20.84 | 20.7 | 20.87 |
| | Highest | 55 | 53.27 | 51.18 | 49.54 | 48.02 | 34.14 |

† In 2014 two surveys were conducted, and the data were pooled together.

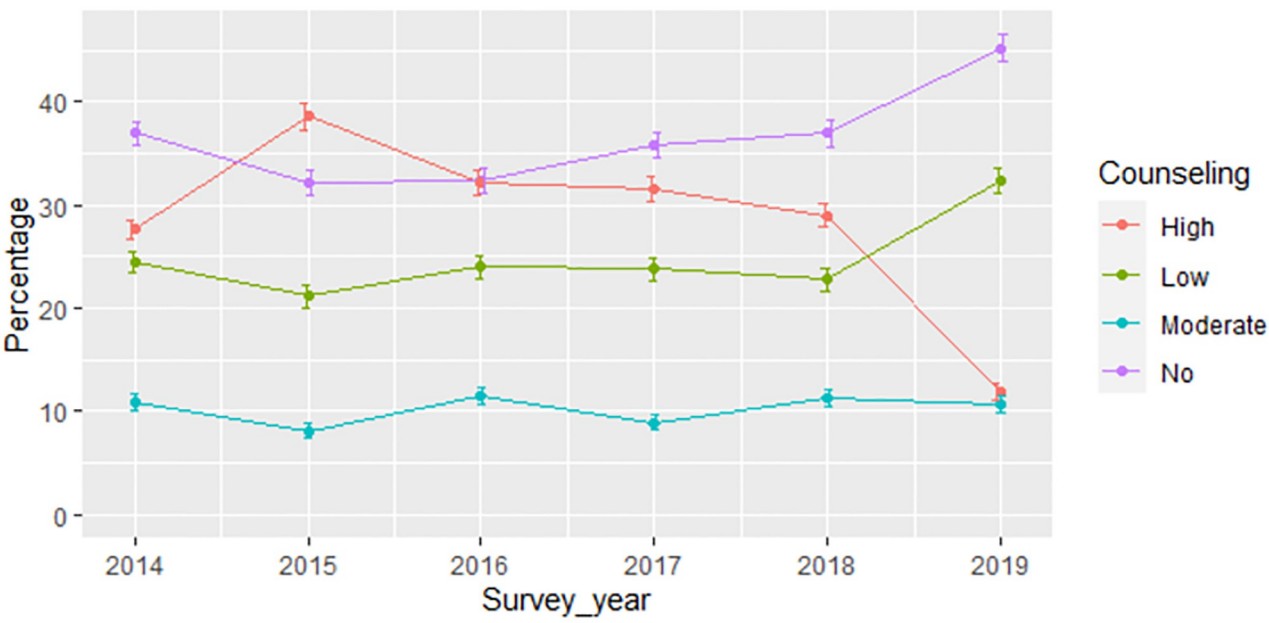

**Fig 1. Percentage of women receiving different levels of family planning counseling service over survey years.**

reported hearing a message about family planning from the media decreased across survey years.

The overall weighted prevalence of receiving high quality family planning counseling service was 28.61% (with 95% CI: 27.68%—29.55%). Table A1 in S1 Appendix presents the percentage distribution of women who received each level of family planning counseling quality by different factors and over the survey-rounds. The percentage of women who received high quality family planning counseling service was 27.6% in 2014, increased to 38.6% in 2015, and thereafter declined to a low of 11.8% in 2019 (Table A1 in S1 Appendix, Fig 1). Receiving a method from a health center or health post, using a long-acting method, and having media exposure to family planning were associated with a higher prevalence of receiving high quality family planning counseling service across all survey rounds.

Table 3 presents a summary of the percentage change in receiving high quality family planning counseling service. Compared with 2018, there was a significant decline (by 59.4%) in the coverage of providing high counseling service in 2019. Overall, the percentage of the women who received high quality family planning counseling service declined between 2014 and 2019 by 57.3 percentage points.

**Table 3. Percentage change in receipt of high-quality family planning services across survey rounds.**

| Base year | Survey year | | | | |
|---|---|---|---|---|---|
| | **2015** | **2016** | **2017** | **2018** | **2019** |
| 2014 | 39.9(19.57, 60.25) | 16.6(-3.03, 36.2) | 14.3(-8.52, 37.2) | 5.1(-16.1, 26.3) | -57.3(-67.11, -47.48) |
| 2015 | | -16.7(-28.10, -5.27) | -18.3(-34.7, -1.9) | -24(-40.1, 9.7) | -69.47(-76.5, -62.4) |
| 2016 | | | -2.0(22.02, 18.17) | -9.84(-28.44, 8.7) | -63.4(-71.9, -54.8) |
| 2017 | | | | -8.1(-21.84, 4.69) | -62.65(-71.05, -54.25) |
| 2018 | | | | | -59.4(-68.56, -50.18) |

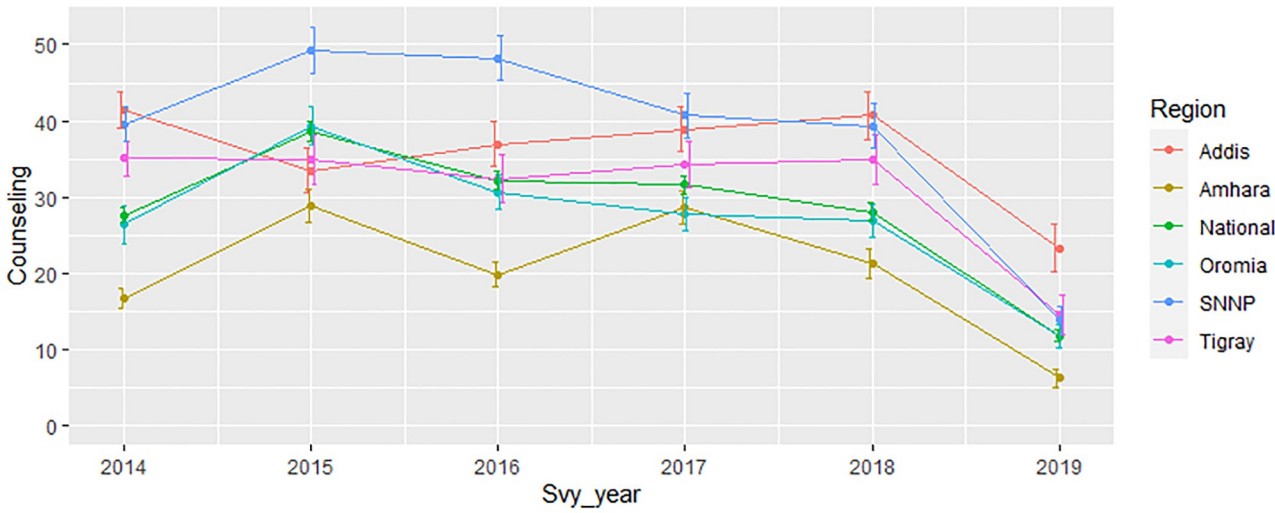

**Fig 2. Percentage of women receiving high quality family planning counseling service over survey years by region.**

The percentage of high quality family planning counseling changed over time with different rate across regions of the country (Fig 2, Table A1 in S1 Appendix). The coverage in the receipt of high quality of counseling in Tigray, Addis Ababa and SSNP region was above the national level while, the coverage was found to be lower in Amhara region (Fig 2).

Fig 3 presents overall trends in the percentage of women who received high quality family planning counseling service across survey year by different factors. The results reveal that, irrespective of the factors examineds, the quality of counseling increased from the year 2014 to 2015 and start declining thereafter.

## 3.2 Modeling results

Table 4 presents the multilevel modeling results by considering one factor at a time (Model A) and after controlling for observed explanatory variables (Model B). Conditional on EA level random effects, the odds of getting counseling service varied across categories for parity, method type, education, media exposure, method source, region of residence, and survey year.

Women who reported being exposed to family planning information through the media, those who received their family planning method from health post or health center, women who attained secondary or higher education, and women using long-acting methods were significantly more likely to receive high counseling service on family planning (Table 4).

Among women with the same random effect value, relative to women who were using short-acting methods, women who were using long-acting methods had 1.90 times the estimated odds of getting "high" counseling service instead of "moderate" or "low" or "no" counseling and the estimated odds of getting "high" or "moderate" counseling service instead of "low" or "no" counseling service and the estimated odds of getting "high" or "moderate" or "low" counseling service instead of "no" counseling service. Among women with the same random effect, compared to women who got their method from a hospital, women who got their method from a health post had 2.18 times the estimated odds of getting "high" counseling service instead of "moderate" or "low" or "no" counseling service and the estimated odds of getting "high" or "moderate" counseling service instead of "low" or "no" counseling service and the estimated odds of getting "high" or "moderate" or "low" counseling service instead of "no" counseling service (Table 4).

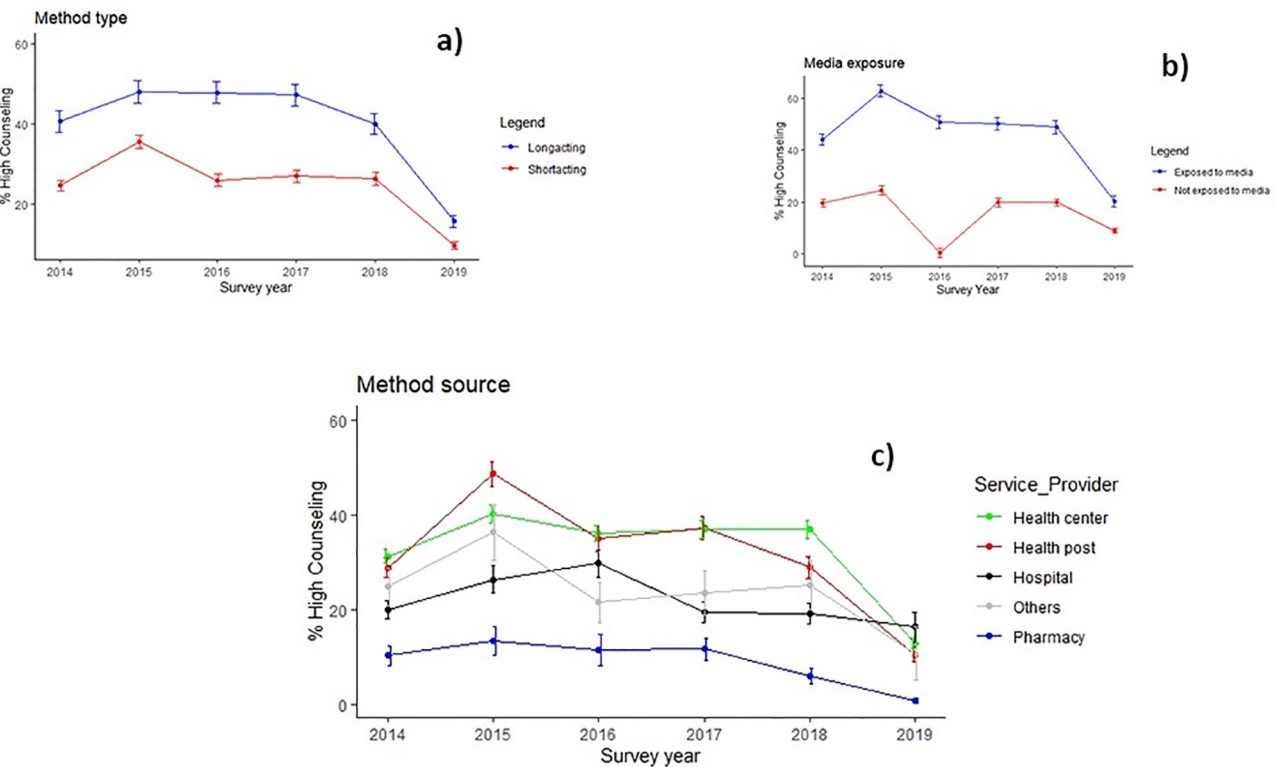

**Fig 3. Trends in receipt of high quality family planning counseling by different factors.** Percentage of women who received high family planning counseling service according to: a) type of method used, b) exposure to family Planning via media, and c) source of contraceptive method.

For Model B (Table 4), the variance partition coefficient (VPC) is $\frac{0.58}{0.58+\frac{\pi^2}{3}} = 0.176$, which indicates that 17.63% if the variance in counseling service can be attributed to differences between EA's.

## 4 Discussion

This study examined factors associated with receipt of high quality family planning counseling services and its trend in Ethiopia from 2014 to 2019. We found that the percentage of women who received high-quality services was low throughout the study period, reaching a high of 38.6% in 2015 and declining sharply to 11.8% in 2019. Significant variation in receipt of high quality counseling was found at the client-level, service delivery point-level, and by region of residence. Nationally, our finding on the trend analysis in the percentage of women receiving high quality family planning counseling service declined overtime. This findings point out some of the gains in improving service coverage were not complimented with improved service quality. In addition, the fact that exposure to family planning message from the media and health professionals has decreased over the years indicate perhaps less attention was given to maintaining and improving the existing public awareness on family planning methods. While the level of high-quality counseling did not exceed 40% at any point, there was a sharp decline in 2019, falling almost 60% from the year before. Among client-level factors, parity, education, media exposure, and type of contraceptive method were found to be statistically significantly associated with high quality family planning counseling services.

Except in Addis Ababa, high-quality family planning counseling peaked in 2015 in all other regions and declined in the following years. The observed relatively better family planning

**Table 4. Parameter estimates from the multilevel ordinal logistic regression model of receiving high quality family planning counseling service in Ethiopia.**

| Factor | Model A | | Model B | |
|---|---|---|---|---|
| | COR | COR 95% CI | AOR | AOR 95% CI |
| Age (ref:15–19) | | | | |
| 20–24 | 1.31 | (1.11, 1.54) | 1.01 | (0.80, 1.24) |
| 25–34 | 1.64 | (1.39, 1.93) | 1.19 | (0.93, 1.53) |
| 35–49 | 1.65 | (1.37, 1.98) | 1.18 | (0.90, 1.56) |
| Wealth Quantile (ref: Lowest) | | | | |
| Lower | 0.88 | (0.74, 1.05) | 0.89 | (0.74, 1.07) |
| Middle | 1.02 | (0.85, 1.22) | 1.03 | (0.84, 1.26) |
| Higher | 1.08 | (0.90, 1.30) | 1.05 | (0.85, 1.29) |
| Highest | 1.23 | (1.01, 1.49) | 1.03 | (0.80, 1.33) |
| Parity (ref: 0 children) | | | | |
| 1- 2 children | 1.77 | (1.56, 2.01) | 1.36 | (1.14, 1.63) |
| 3+ children | 1.89 | (1.65, 2.17) | 1.47 | (1.16, 2.14) |
| Method type (ref: short acting) | | | | |
| Long-acting | 2.29 | (2.04, 2.57) | 1.90 | (1.69, 2.14) |
| Education (ref: Never attended) | | | | |
| Primary | 0.99 | (0.89, 1.09) | 1.07 | (0.95, 1.21) |
| Secondary/higher | 1.16 | (1.02, 1.32) | 1.38 | (1.17, 1.62) |
| Media exposure (Ref: No) | | | | |
| Yes | 1.75 | (1.59, 1.93) | 1.68 | (1.51, 1.88) |
| Method source (ref: Hospital) | | | | |
| Health post | 1.82 | (1.53, 2.16) | 2.18 | (1.80, 2.63) |
| Health center | 2.02 | (1.74, 2.36) | 2.15 | (1.82, 2.54) |
| Pharmacy | 0.23 | (0.18, 0.30) | 0.29 | (0.22, 0.38) |
| Other | 1.01 | (0.76, 1.35) | 1.09 | (0.83, 1.42) |
| Residence (ref: Urban) | | | | |
| Rural | 0.84 | (0.72, 0.98) | 0.99 | (0.80, 1.22) |
| Region (ref: Addis Ababa) | | | | |
| Tigray | 0.86 | (0.65, 1.13) | 0.78 | (0.61, 0.99) |
| Amhara | 0.44 | (0.34, 0.57) | 0.47 | (0.37, 0.59) |
| Oromiya | 0.69 | (0.53, 0.89) | 0.67 | (0.53, 0.84) |
| SNNP | 0.92 | (0.71, 1.19) | 0.88 | (0.69, 1.13) |
| Other | 0.69 | (0.51, 0.93) | 0.75 | (0.53, 1.07) |
| Survey year (ref:2014) | | | | |
| 2015 | 1.41 | (1.10, 1.79) | 1.53 | (1.15, 2.03) |
| 2016 | 1.26 | (1.02, 1.56) | 1.25 | (1.01, 1.55) |
| 2017 | 1.07 | (0.84, 1.35) | 1.11 | (0.86, 1.42) |
| 2018 | 0.98 | (0.79, 1.23) | 1.02 | (0.81, 1.27) |
| 2019 | 0.54 | (0.44, 0.67) | 0.43 | (0.35, 0.54) |
| **Random effect** | | | | |
| Var(EA) | 0.71 | | 0.58 | (0.47, 0.73) |
| VPC | 21.58% | | 17.63% | |

Note: Model A is fitted by considering only one predictor variable at time, and the variance of the random intercept varies from 0.60 to 0.81 with an average variance of 0.71. Model B is fitted by adjusting for client-level, service provider-level, and place of residence variables simultaneously. VPC represents the percentage of total variation of receiving high family planning counseling attributable by EA

counseling service in 2015 may be due to the fact the Ethiopian health sector transformation plan started in this year [31]. As compared with other regions, high quality counseling service was low in Amhara region and sharply declined since 2017 (Fig 2). This aligns with Hrusa's finding using PMA data [12].

At the client-level, Jain found that education and wealth were generally associated with higher quality services [8]. We found no differences by wealth, though we did see that women who attended secondary or higher education were significantly more likely to report receiving higher quality services than women without education. Similarly, women who heard about family planning in the media were also more likely to indicate they received higher quality counseling services. These results provide support for Jain's hypothesis that women with more education may have greater method knowledge and greater ability to recall the information exchanged during their family planning visit. Engagement with family planning media may also lead to greater knowledge of different contraceptive methods and their side effects, allowing women to ask questions of providers that may otherwise be overlooked. Continued training to ensure that comprehensive contraceptive information is delivered in simple, understandable language may alleviate some educational disparities, as would investment in ongoing family planning media campaigns. Messaging that is developed in conjunction with regional media, and distributed in local languages, may support greater general knowledge of family planning and support women in asking questions and receiving high quality services. Media exposure enables a more informative client-provider interaction because it's enhances client knowledge and empowers them to be more inquisitive in the family planning visit.

In line with findings in other countries, education, parity have impact on receiving high family planning counseling service [8, 15, 32–34]. A study by [14] showed, 58.9% of the counseling sessions did not maintain the privacy of clients during the consultation and 74.2% of them were not told about the possible side effects related to the use of a method. A study done in the northern part of Ethiopia shows that only 22.4% of the clients chose contraceptive method by themselves and 74.1% of them with assistance of the provider [13]. Variation in access to and quality of family planning services by parity and method-type has been documented extensively across settings and has identified consistent barriers to high quality family planning services among unmarried adolescents, including provider bias and lack of training [14, 35, 36]. Policies and programs designed to expand and strengthen youth friendly services must emphasize that the delivery of comprehensive information is a critical component of high-quality youth friendly services, in addition to improving privacy, confidentiality, and reducing provider bias.

Finally, women who reported using a long-acting method were significantly more likely to report receiving high quality services than women using short-acting methods. This aligns with Jain's findings that the MII was highest among implant users, followed by the IUD [8]. While it is positive that women who choose long-acting methods are made aware of potential side effects and how to address them, the majority of contraceptive users in Ethiopia use the injectable [9]. Even as the method mix changes in Ethiopia, with greater use of long-acting methods, it is important to ensure that women who continue to prefer short-acting methods are given comprehensive information.

At the service provider-level, method source was significantly associated with quality of family planning counseling service. Women who received care from pharmacies had significantly lower odds of receiveing high-quality family planning counseling than women who received care at either a health post or health center. In addition to differences in service modality and training of service providers, this likely reflects differences in characteristics of users and method choice; other studies have shown that women who use pharmacies tend to

live in urban areas, be younger, and rely on short-term methods such as emergency contraceptives and condoms which are typically not preceded by sufficient counseling [37, 38].

We did not find any differences by urban or rural residence, which is contrary to the recent Hrusa study in Ethiopia that found rural women were more likely to receive high quality services [12]. This may be due to the fact that the paper did not account for clustering at the EA level, which may be particularly important among rural women who have fewer health facilities available to them than urban women. While Jain found that across 25 countries, urban women were more likely to receive high quality counseling, Ethiopia has invested heavily in its public health system, particularly in rural areas, to address disparities in access to and use of family planning services [31]. This, combined with the generally higher level of services available at public health facilities (i.e., health post and health centers), may explain why urban and rural differences are not pronounced.

The results presented in this study should be considered in light of some limitations. One limitation is the fact that the MII does not capture the full range of elements necessary to measure high-quality contraceptive counseling. Additionally, due to the nature of the household survey which asks women to recall information about the first time they received their current method, there may be a recall bias. Further, covariates considered in this study were limited to those available in the survey which lacks supply-side variables such as availability of methods at facilities, availability of trained staff, job aids, information leaflets for clients and other facility level variables. Since the analysis results in this study were derived from cross-sectional female and household surveys, and quality of counseling is heavily influenced by facility-level factors, future studies should consider using PMA service delivery point (SDP) surveys.

## 5 Conclusions and recommendation

Quality of family planning services is low in Ethiopia, with evidence of a sharp decline in recent years. Significant disparities by region, method source and by socio-demographic characteristics exist, particularly among uneducated women, media exposure about family planning and by choice of method. Given the importance of continuous provision of information on the range of family planning methods and their side effects, it is imperative to develop simple, culturally appropriate messages in local languages and collaborate with regional media as an important vehicle of information on family planning. Interventions aimed at improving quality of family planning counseling need to be mindful of the regional disparities and ensure equity in service quality. Further, the results of this study reveal that high-quality family planning counseling service tended to be more available at health posts and health centers compared to hospitals and is particularly poor at pharmacies. Thus, the Ministry of Health and partners should support training on contraceptive counseling for pharmacists and related auxiliary health staff to improve the information provided through the private sector.

## Supporting information

**S1 Appendix.**
(PDF)

## Acknowledgments

The authors greatly appreciate the comments of Colin Baynes and Orvalho Augusto. The paper has been considerably strengthened due to their suggestions.

## Author Contributions

**Conceptualization:** Bedilu Alamirie Ejigu, Solomon Shiferaw.

**Data curation:** Bedilu Alamirie Ejigu.

**Formal analysis:** Bedilu Alamirie Ejigu.

**Funding acquisition:** Linnea Zimmerman.

**Supervision:** Solomon Shiferaw.

**Validation:** Assefa Seme, Solomon Shiferaw.

**Writing – original draft:** Bedilu Alamirie Ejigu.

**Writing – review & editing:** Bedilu Alamirie Ejigu, Assefa Seme, Linnea Zimmerman, Solomon Shiferaw.

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
