## [Decision Letter · Decision Letter 0]

31 Aug 2021

PONE-D-21-19351

Trend and Determinants of Quality of Family Planning Counseling in Ethiopia: Evidence from repeated PMA cross-sectional surveys, (2014-2019)

PLOS ONE

Dear Dr. Ejigu,

Thank you for submitting your manuscript to PLOS ONE. After careful consideration, we feel that it has merit but does not fully meet PLOS ONE’s publication criteria as it currently stands. Therefore, we invite you to submit a revised version of the manuscript that addresses the points raised during the review process.

We look forward to receiving your revised manuscript.

Kind regards,

Orvalho Augusto, MD, MPH

Academic Editor

PLOS ONE

Journal Requirements:

"This study was conducted with support received from the Bill & Melinda Gates Foundation through a grant received by the Bill & Melinda Gates Institute for 

Population and Reproductive Health for the Performance Monitoring for Action Ethiopia (PMA-Ethiopia) projects."

"This study was conducted with support

received from the Bill & Melinda Gates

Foundation through a grant received by the Bill & Melinda Gates Institute for

Population and Reproductive Health for the Performance Monitoring for Action

Ethiopia (PMA-Ethiopia) projects."

Additional Editor Comments (if provided):

Family planning is such a very important service. Here the authors report a longitudinal analysis of quality counselling using PMA surveys. I applaud the authors for first, use longitudinally a series of 7 cross-sectional datasets, secondly for defining just one ordinal outcome rather than making 3 logistic regression (as many would do). This is a very efficient use of data. However, it is still debatable to use just these 3 variables as indicators of the quality of counselling. I see no discussion of this.

Issues

I. Abstract:

Please split this into introduction, methods, results and conclusions

II. Introduction

This is a very good background. Just a clarification on line 12 what are these numbers (0.77 to 3.17)? Are these yearly changes?

III. Methods

1. In the “study design and data source” we are pointed to reference 26 to learn the sampling details of PMA2020. That is fine but brief details of such sampling should be added here. It would be great if details of what is the size of an Enumeration Area or of the PSU relative to an administrative area in Ethiopia. How is comparable?

2. What is this method of information index? (lines 77/78)

3. Please add spaces at “if a woman got allthethree” between lines 84 and 85

4. Equation 2 - nowhere is explained the meaning of sigma2 squared (the variance of the random-effects).

5. Table 1 - please either specify what “svy-year” means or write “survey year”.

6. Line 122 - the Brant is a name. So please correct it to be “Brant test”.

7. Line 136 - Stata is not an acronym. So correct STATA to Stata please. See the official documentation for reference. For example, your reference 34 clarifies that.

8. Line 137, I suggest adding the reference for ggplot2. Your plots are based on ggplot2. Correct?

9. Was any goodness of fitness performed in this analysis?

IV. Results

1. What is the amount of missing data here?

2. Table 2 please on the caption/footnote that these are proportions in percentages.

- Are these weighted proportions?

- all percentages seem to respect the 2 decimal places. So why not “2014 FP message” “2015 FP Facility”?

3. Table 3 there are empty cells. Such as 2019 other regions. Can you check this, please?

4. Line 166 no figure or table indicated.

5. How the confidence intervals for the relative changes are computed?

6. Table 5 - What are COR and AOR? Put in the footnote, please. Here should be the place to add the variance component proportion. In fact, I do not see this in the results.

V. Discussion:

Why no limitations discussion here?

Reviewers' comments:

Reviewer's Responses to Questions

**Comments to the Author**

1. Is the manuscript technically sound, and do the data support the conclusions?

Reviewer #1: No

2. Has the statistical analysis been performed appropriately and rigorously? 

Reviewer #1: No

3. Have the authors made all data underlying the findings in their manuscript fully available?

Reviewer #1: No

4. Is the manuscript presented in an intelligible fashion and written in standard English?

Reviewer #1: No

5. Review Comments to the Author

Reviewer #1: The author chose a very important topic and I admire her/his statistical knowledge and skills. I would very much like to see a paper emerge on this subject. However, I feel that the authors have missed the mark in several respects that I outline below. I recommend that the authors take my feedback into consideration and determine how best to proceed themselves. The description of the data was so inadequate that I would not have been able to make my way through the paper had I not been somewhat familiar with the data source myself; however, my feedback might have missed the mark also since I had to make assumptions about the data source. Still, I would not reject a paper on this topic with these data, taking into account what else I know about PMA 2020 data but I think the analysis presented here with cannot go forward without first presenting the data source and collection methods more clearly, more thoroughly conceptualizing the notion of FP counseling quality in the wider context of FP service quality and mapping the factors that influence it vis-à-vis the array of indicators in the data sources available, and then thinking through more soundly what factors are most feasible and relevant for their analysis given the data at hand given the possibility of biases.

6. PLOS authors have the option to publish the peer review history of their article (what does this mean?). If published, this will include your full peer review and any attached files.

Reviewer #1: **Yes: **Colin Baynes

---

## [Author Response · Author response to Decision Letter 0]

2 Dec 2021

Response to Editor and Reviewers

Editor Comments 

Family planning is such a very important service. Here the authors report a longitudinal analysis of quality counselling using PMA surveys. I applaud the authors for first, use longitudinally a series of 7 cross-sectional datasets, secondly for defining just one ordinal outcome rather than making 3 logistic regressions (as many would do). This is a very efficient use of data. However, it is still debatable to use just these 3 variables as indicators of the quality of counselling. I see no discussion of this.

Issues

I. Abstract:

Please split this into introduction, methods, results and conclusions

 The abstract section updated accordingly.

II. Introduction

This is a very good background. Just a clarification on line 12 what are these numbers (0.77 to 3.17)? Are these yearly changes?

- It is an annual rate of changes (percentage points) in modern contraceptive prevalence rates among all women of reproductive age (15–49 years).

III. Methods

1. In the “study design and data source” we are pointed to reference 26 to learn the sampling details of PMA2020. That is fine but brief details of such sampling should be added here. It would be great if details of what is the size of an Enumeration Area or of the PSU relative to an administrative area in Ethiopia. How is comparable?

 - In the revised version of the manuscript, additional clarification in this issue was given.

2. What is this method of information index? (lines 77/78)

 - It is a composite score for family planning service quality.

3. Please add spaces at “if a woman got allthethree” between lines 84 and 85

 - Thanks for pointing this, updated accordingly.

4. Equation 2 - nowhere is explained the meaning of sigma2 squared (the variance of the random-effects).

- The variance of the random effect (\\sigma_{2}^{2}) represents how much variability there is between clusters with respect to the response variable. 

5. Table 1 - please either specify what “svy-year” means or write “survey year”.

 Updated.

6. Line 122 - the Brant is a name. So please correct it to be “Brant test”.

 - Corrected.

7. Line 136 - Stata is not an acronym. So correct STATA to Stata please. See the official documentation for reference. For example, your reference 34 clarifies that.

 - Corrected.

8. Line 137, I suggest adding the reference for ggplot2. Your plots are based on ggplot2. Correct?

Yes, and reference on ggplot2 included.

9. Was any goodness of fitness performed in this analysis?

- Yes. Likelihood ratio test was employed to assess the importance of incorporating the random effect term in the model. Further, AIC was used to compare models with different covariates.

IV. Results

1. What is the amount of missing data here?

- Our analytical samples are all women who completed the survey questionnaires. counseling service.

2. Table 2 please on the caption/footnote that these are proportions in percentages.

- Are these weighted proportions? Yes

- all percentages seem to respect the 2 decimal places. So why not “2014 FP message” “2015 FP Facility”? Corrected.

3. Table 3 there are empty cells. Such as 2019 other regions. Can you check this, please?

- Checked and addressed.

4. Line 166 no figure or table indicated.

- Addressed.

5. How the confidence intervals for the relative changes are computed?

- Using the linearization approach.

6. Table 5 - What are COR and AOR? Put in the footnote, please. Here should be the place to add the variance component proportion. In fact, I do not see this in the results.

- COR=Crude Odds Ratio, AOR= Adjusted Odds Ratio

V. Discussion:

Why no limitations discussion here?

- In the revised version of the manuscript, a limitation section included. 

Reviewer #1:

 The author chose a very important topic and I admire her/his statistical knowledge and skills. I would very much like to see a paper emerge on this subject. However, I feel that the authors have missed the mark in several respects that I outline below. I recommend that the authors take my feedback into consideration and determine how best to proceed themselves. The description of the data was so inadequate that I would not have been able to make my way through the paper had I not been somewhat familiar with the data source myself; however, my feedback might have missed the mark also since I had to make assumptions about the data source. Still, I would not reject a paper on this topic with these data, taking into account what else I know about PMA 2020 data but I think the analysis presented here with cannot go forward without first presenting the data source and collection methods more clearly, more thoroughly conceptualizing the notion of FP counseling quality in the wider context of FP service quality and mapping the factors that influence it vis-à-vis the array of indicators in the data sources available, and then thinking through more soundly what factors are most feasible and relevant for their analysis given the data at hand given the possibility of biases.

Abstract

• I would not refer to 2014 as “baseline” and 2019 as “endline” since that implies that there was an intervention in between the years.

• “A multi-level model was used to examine potential factors”. I would state your research question(s) related to exploring trends and associations more clearly and for each the corresponding statistical strategy you use to answer the question.

• What is the operational definition of “good family planning counseling” and “poor family planning counseling”? How did you construct it as a variable in the statistical analyses?

• For readers that do not know indicate briefly the type of data that you are using (i.e., household surveys, facility-based exit interviews).

- In the revised version of the manuscript, the abstract entirely updated based on the Editor and reviewer comments. 

Background

“In sub-Saharan Africa countries the trend in using 10 modern contraceptive method among women of reproductive age varied substantially from 0.77 to 3.17” – I do not understand this sentence. What is meant by 0.77 to 3.17?

- It is an annual rate of changes (percentage points) in modern contraceptive prevalence rates among all women of reproductive age (15–49 years).

“Prevalence of family planning coverage” – What is this? Prevalence to me connotes the level of method use among individuals in a population and coverage implies that amount of population that has access to a service (two different things). Similarly in line 15 you say “coverage” of contraception is associated with individual-level covariates. Do you mean “use of contraception”?

- Yes, to mean “use of contraception”, and updated accordingly.

The author frames the background with the global situation of contraception prevalence and unmet need in broad terms and leaps to a discussion of very focused set of factors that influence method use, counseling. A more useful framing might start by quickly acknowledging gaps in availability and use of FP services among women in sub-Saharan Africa, and proceed to frame the paper around the need to better understand the quality of care – pointing out the multiple aspects of family planning service quality and a overview of the approaches that have been used to measure and evaluate FP service quality, evidence on the determinants and consequences of FP service quality, then introducing the subject of this paper – the determinants of FP counseling quality - in terms of the particular knowledge gap that it is filling.

- We greatly appreciate the feedbacks, and the background section updated.

Be sure to clearly define family planning counseling and how the quality of FP counseling has been defined and studied in the literature. There are multiple dimensions to it and ways of measuring it (depending on whether it is explored from the client or provider perspective, whether the focus is on technical information imparted during the exchange, the rapport between client and provider, the service delivery environment and its privacy and availability of other information such as leaflets, client recall of specific messages about side effects versus their general recall as to whether more than one method was discussed, use of mystery clients, direct observations, client-exit interviews immediately after the service, household surveys that occur much after the service exchange, assessments of provider biases and knowledge, etc.). It is crucial to acknowledge this and describe this paper’s definition of FP counseling quality in that context. The current draft seems to assume that FP counseling quality and information exchange are the same thing when in fact, I think, the latter is an aspect of the former. 

 Then proceed to explain the choice to focus specifically on the “information exchange”, why this is novel and appropriate. Some useful information for this is already in the paper, but it is not clear what dimension of quality and family planning quality that they measured and report on. 

“Studies on family planning counseling service in Ethiopia showed that 40 the coverage is below 40%” – What does this mean exactly?

- Percentage of women who received high quality family planning counseling service is below 40%. The sentence updated in the revised version of the manuscript.

Lines 42-49, make clear the type of data you are talking about when you are citing the literature. Were these household surveys in which women were asked to recall information about the last time they sought FP services? Or were they exit-interviews conducted immediately after women were discharged from care? Did the surveys assess quality from the client or provider perspective (i.e., who did they enroll)? 

- It is a household surveys in which women were asked to recall information about the last time they sought family planning counseling service. So, the survey assesses the quality from client perspective. 

 Materials and Methods

“The PMA surveys are repeated cross-sectional surveys based on a multistage stratified cluster sampling design.” Who do they interview and where? What is the operational definition of the enumeration area” and what is the operational definition of “population proportion to size” for these surveys? What does the term “multistage stratified cluster sampling design” mean for this study specifically (define the stages, define the cluster and strata). Did the sampling frame ever change during the seven years of survey implementation and if so how often? I prefer to see this information in the paper and not refer to another paper. I am familiar enough with PMA 2020 to know that it is well done, but I find this information rather crucial to understand the rest of the manuscript so I think you should provide it. 

- Thanks for the feedback, and updated accordingly.

You refer to “the outcome variable” in line 70 but at this juncture I do not think that you have formally defined it yet. 

- Addressed. 

Variables: I find this section extremely difficult to understand because I do not know if the survey respondents were women of reproductive age enrolled in households, actual clients enrolled immediately after discharge from the service, providers that delivered the service and report of their own performance. Since I am familiar with PMA 2020 I assume you mean WRA enrolled at the household, so I can make it through the manuscript. But describe the sample clearly and how they are distinguished from the underlying population. Presumably your sample is a sub-sample of women from the surveys that had received FP services within some time frame before the survey, or are they simply reporting on the last time they ever sought FP services? Are they all FP users or acceptors or just anyone that received any counseling? Clarify the range of the duration representing the time between when they index counseling interaction took place and the time of the interview.

- Thanks for this comment. In the revised manuscript, study samples were clearly described. 

Table 1: The third age category seems quite broad. I suggest breaking it down to 25-34, and 35+. What about permanent methods. Are you able to find out if they are new or return method users? Media exposure: do you mean exposure to media before or after the service episode. This may be a minor point, but consider whether it is relevant that the client-level factors represent the status of the client at the time of the interview and not when they obtained family planning.

- We categorized age as adolescent, young and adult for interpretation purpose. In the revised version we consider your comments, and recategorized age as: 15-19, 20-24, 25-34 and 35-49.

- Media exposure before the service episode.

I encourage the author to map out a causal pathway model between each proposed “exposure” and the “outcome” (quality of information exchange) taking into consideration the possibility of the exposure occurring after the dependent variable is mentioned, the theoretical linkages between the exposures and the quality of FP counseling, and possible recall bias. Its not necessary to include this in the paper, but the author may find it beneficial to do so and it is up to her/him.

In this vein, some issues to consider:

- Some of the client level factors may predispose clients to be more or less assertive during the interaction with provider (e.g., ask questions, request clarifications, inquire about different methods they have heard of). In this case, would the outcome reflect variation in the quality of counseling or rather the common sense that information exchange is better when clients ask questions.

- Some of the client level factors predispose some clients to poorer recall of counseling information than others.

- I am concerned about “FP discussion” as a exploratory variable (presumably you mean independent variable, possibly a predictor of interest)? How is an “FP discussion” different from the information exchange whose rating is treated as the outcome? Is this not problematic for your model? 

- Thanks for raising this issue. We acknowledge your comment, and in the revised manuscript, we drop this variable from our analysis/model.

- I would include type of facility since I know they are many levels of care in the Ethiopian healthcare system (e.g., health post, health center, hospital, pharmacy, community based distributor). 

- Thanks again. Yes! We did it now. 

- Also it makes a difference if the woman is receiving a re-injection or refilling a prescription of oral contraceptives (she may not need counseling since she has been on the method for some time anyway, and I imagine you’ll find that counseling quality is lower for short-term methods for that reason). 

- Yes, that is true.

- Can you include cadre type in the model (e.g. health extension worker, nurse, physician)? 

- The point you raised is interesting, but we don’t have this data in this survey. 

- The satisfaction variable seems problematic to me for the same reasons that I pointed out with respect to the “FP discussion” – it seems like a dimension of FP counseling quality in the same way that the information exchange is and I do not see how one can sensibly include in both sides of the equation.

- Thanks for raising this issue. We acknowledge your comment, and drop this variable from our analysis/model.

- Some one that received information on contraceptive method from a provider, especially a user of a method, may be more likely to report media exposure that occurred after the service because they were predisposed to being interested in family planning.

- Similarly exposure to media before the service may prompt women to seek family planning services and ask questions during the exchange with the provider. Thus, (1) there may be the problem of reverse causality, and (2) does this really tell us anything about the quality of counseling or rather the effect of media exposure on recall, health behaviors and assertiveness of client during exchanges with provider? 

- Having media exposure on family planning may help the client to seek more information on the three key question (MII defined) from the provider which ultimately improve the quality of counseling.

- As you think through the causal pathway model, consider other factors such as whether the facility had commodities and supplies when the client sought FP services, whether the provider had ever received FP training and when.

- Thanks for the feedback. In the future, we will consider to collect such data in the upcoming surveys.

Statistcal analysis

Describe in greater detail how the “data analysis methods considered sampling weights for generating unbiased population estimates” and why the authors believed that this was necessary in the multi-level model with covariates? How did you incorporate the design-based approach to address unequal sampling probabilities in the multilevel model? Reading farther into the manuscript I see that you (might) have gone through the weighted procedures to obtain the descriptive results, but this also needs to be explained more clearly. Make very clear what were the different components of your analysis and the specific methods you implemented for each respectively. Generally, the lack of clarity in this manuscript has made reviewing this article time consuming and difficult. The authors should have their work reviewed by peers that can give this feedback before submitting to journals.

Include a statement that in as complete and plainspoken terms as possible gives the interpretation of the relationship between your independent and dependent variables that your analysis seeks and how the POM model ascertains this. The application of the POM to longitudinal data with repeat cross-sections is novel and interesting, but complex and readers should be guided in understanding how to interpret that aspect of the model as well. But, a reader that has never done ordered logistic regression but is otherwise conversant in regression will not be able to understand what you are saying. If I had not recently done a similar analysis myself I would be very confused.

Did you apply the Brant test to both models or just model 2?

- In both models. But now, since we drop FP discussion from our model, and re-categorize method source as Hospital, health-center, health-post, pharmacy and other, the proportional odds assumption satisfied.

- 

Results

Table 3 is dizzying. I suggest you innovate with plots to convey the most important trends through a series of carefully tailored visualization, and elucidate these with parsimonious commentary and include this table as a Appendix only. 

- We considered your suggestion and updated accordingly.

I think you need to reconsider the findings in light of some of the comments I made above about recall biases, counseling for STM v. LARC, reverse causation in association between media exposure and the outcome and what this tells us about counseling quality, the appropriateness of “FP discussion” and “satisfaction” as predictors, and absent data on the readiness for FP service delivery (available methods in stock, trained providers). In my opinion, the findings section needs to be strengthened quite appreciably to withstand that critique.

Your explanation of the model results does not reflect the ordered nature of the response variable. How do you interpret the OR in terms of the predictor response association reported by the POM?

- Thanks for pointing this. Updated.

Discussion

I suspect that this section may change significantly upon reflection of my comments above. The trend in the quality of the information exchange during FP counseling is noteworthy and merits elucidation in a published paper, but it seems that this analysis has missed the mark. 

There is no question that provider bias is at play, but is it really fair to point to this as the driver of the national trend of declining FP counseling quality when the analysis does not acknowledge anything about the providers’ background, training, working environment, access to commodities and appropriate supervision? 

Are LARC available in the private sector and if so is this restricted to particulate social franchises? Or are they mostly available in the public sector. It makes sense that LARC provision is associated with associated with more information provision since they entail removal services (unlike oral and injectable methods) and often times these occur in public sector settings. This does not mean that public sector providers are better counselors but the specialize in services that oblige them to share more information. 

- As you suggested above, in the revised version of the manuscript, rather than public Vs private, we categorize the method source by facility type.

The issue of recall bias needs to be addressed squarely acknowledging the possibility that respondents may conflate the information they receive about FP methods from their provider with FP information that they heard from another source either before or after they had received FP services. 

 - Yes, that is true and we mentioned this fact under the limitation Section of the manuscript.

I think a more interesting analysis would focus on the individual characteristics of the provider (training, age, job title, sex, etc.), working environment (facility type, access to commodities, supervision, etc.), and macro-context (region, etc.). Pinpointing the quality of information that comes from provider to the client on characteristics of the client raises to the fore issues of provider bias that we already know about but cannot disentangle from the possibility that women with certain characteristics are more likely to have better recall than others. 

- We agree with your suggestions, but in our survey considered for this analysis, we don’t have data about the individual characteristics of the provider linked with each client.

---

## [Decision Letter · Decision Letter 1]

17 Jan 2022

PONE-D-21-19351R1Trend and Determinants of Quality of Family Planning Counseling in Ethiopia: Evidence from repeated PMA cross-sectional surveys, (2014-2019)PLOS ONE

Dear Dr. Ejigu,

Thank you for submitting your manuscript to PLOS ONE. After careful consideration, we feel that it has merit but does not fully meet PLOS ONE’s publication criteria as it currently stands. Therefore, we invite you to submit a revised version of the manuscript that addresses the points raised during the review process.

We look forward to receiving your revised manuscript.

Kind regards,

Orvalho Augusto, MD, MPH

Academic Editor

PLOS ONE

Journal Requirements:

Additional Editor Comments:

This is an important analysis as I expressed in the previous iterations. The manuscript has improved and the authors addressed most of my comments/questions.

My previous question remaining or with unsatisfactory response:

1. I did ask about the missingness before. Clarifying, what is the response rate of these PMA surveys? What was done if there were participants with missing data among the variables included for this analysis? This could be placed between the current lines 77 to 83.

New few issues:

1. The reviewer below, points out the need for some introduction of the concept family planning (FP) service quality and how that ends be measured as just 3 questions.

2. Line21: What is this coverage? Is this high quality of FP coverage?

3. Line 63 : the SNNP abbreviation is used for the first time here. Can you write fully what it means?

4. Just for clarification: was the year of survey included in the model as a categorical variable. Correct? Can you clarify that in table 1.

5. Results related to table 2. Can you clarify how the weights of the two surveys were combined for 2014 analysis?

6. Lines 159: what were the conclusions of the Brant tests? Was the PO assumption kept across all variables?

7. The paragraph 171 to 174 states that all analysis used the sample weights. This leads to 2 questions:

a. How multiple survey weights were combined for this analysis?

b. It is very tricky to do random-effects models with survey weights. How the results would differ if you did not use weights. This could be placed in the supplements if indeed you did a random-effects model with sample weights.

8. The limitations should be part of the discussion as the reviewer points out below. Somewhere the last 3 paragraphs would be OK.

9. Thank you for the good plots. Just few things:

a. All of them should have the vertical y-axis start at zero. In ggplot2 there is something like expand_limits(y = 0) to sort that.

b. Make all of them to have the same maximum. It is easy to misunderstand the plots as they have different scales.

Reviewers' comments:

Reviewer's Responses to Questions

**Comments to the Author**

1. If the authors have adequately addressed your comments raised in a previous round of review and you feel that this manuscript is now acceptable for publication, you may indicate that here to bypass the “Comments to the Author” section, enter your conflict of interest statement in the “Confidential to Editor” section, and submit your "Accept" recommendation.

Reviewer #1: (No Response)

2. Is the manuscript technically sound, and do the data support the conclusions?

Reviewer #1: No

3. Has the statistical analysis been performed appropriately and rigorously? 

Reviewer #1: Yes

4. Have the authors made all data underlying the findings in their manuscript fully available?

Reviewer #1: Yes

5. Is the manuscript presented in an intelligible fashion and written in standard English?

Reviewer #1: Yes

6. Review Comments to the Author

Reviewer #1: Abstract

In methods section it is written “women aged 1549” but it should say “15 to 49”.

In results section what is meant by “the percentage of women that received high family planning counseling service…” Do you mean high quality FP services?

I am not sure if I use the term “determinants” to characterize the factors that you found were associated with FP counseling quality. For example, I think media is a contextual factor that may be associated with experiencing better counseling quality, but I question whether it truly determines the level of counseling quality.

Even though this may cause the authors to exceed the word limit for the abstract, I think the abstract would be strengthened if it briefly stated how the authors define FP counseling quality.

Background

The new paragraph that points out that in a study on MII across 25 countries “residence, education, household wealth and method type were generally found to be associated with receiving higher quality care”… To be helpful, please specify residence where and what method types were associated with higher quality care.

In the same paragraph, when you say “individual level factors associated with receiving high quality care” it would be useful to clarify that you mean client-level (as opposed to individual provider-level).

Materials and Methods

Are you able to account for whether or not the survey participants were new or returning users? This matters since it is possible that returning users had already received the counseling information in recent FP visits at their nearby facility.

To be helpful, specify what methods you define as long-acting and short-acting.

Results

“More than 70% of women in all years got received their contraceptive methods…” Check for grammar.

Line 258 on page 8: you mention a decline in the proportion of survey participants that report good counseling services, but you say it is a decline in coverage. Elsewhere you call it percentage of women received high quality FP counseling. I question whether coverage is the best term for this. I think it is a measure of the prevalence. Please be consistent in how you refer to the measure.

Line 262 on page 8 references Table?? – please correct this.

Lines 255-267 page 9 – please check to see if you have used the correct tense.

Figure 2 label- “trends in getting high family planning counseling…” should it not say high quality family planning?

I understand the purpose of Table 4, but it is incredibly confusing. I suggest a table with a row for every year in the time series and columns for year, % change between successive years including the 95% CI in parentheses and cumulative % change between current and start year with 95% CI in parentheses.

Lines 278-286- there seems to be some repetition here… please check and revise accordingly.

Thanks for explaining the interpretation of Models A and B. After providing a thorough explanation of the interpretation for one of the explanatory variables, I do not think you need to repeat the same language over again for the other variables. But it is up to you.

In your discussion, you mention the statistical significance of certain associations in your models. Can you include this in Table 5. No need to put the actual p-value but clarify the alpha thresholds you are using (e.g., 0.05, 0.01, 0.0001) and whether the associations are above or below using asterisks.

Discussion

It would be interesting if the authors could elaborate further on what might be the drivers of the decline in the outcome variable over the span of the time series.

I think it is good that in the discussion you mention that the association you identified between media exposure and the outcome is likely due to the fact that the former enhances knowledge and may prompt clients to ask more questions. In link 382 of page 15 you say this supports women in receiving high quality services, but I think a better way of phrasing it is that media exposure enables a more informative client-provider interaction because it’s enhances client knowledge and empowers them to more inquisitive in the FP visit.

Can you establish whether women’s reports of FP media exposure occurred before their reported FP visits in each survey? If not, then you should acknowledge that since women that received FP services may be more likely to seek out or recall FP-related media exposure (e.g., the visit peaked their interest, women that listen to educational media programs are more educated and therefore also more likely to use FP and ask questions during the visit). In other words, the direction of the association between media exposure and FP quality cannot be determined by this analysis, which limits the interpretation.

The abstract mentions that seeking FP counseling from pharmacies was associated with lower levels of FP counseling quality. This is an interesting finding and I think the authors should elaborate on this further in the discussion.

In the discussion, you mention that the odds of receiving high quality FP counseling are higher among women that obtain the service at public facilities compared to those that sought it at private facilities and refer to table 5 (lines 393-397 on page 16). But Table 5 does not include columns that refer to public vs. private facilities. Table 5 does give parameter estimates for health post, health center, pharmacy and other facility (ref hospital). This finding could be elaborated on in the discussion, with some clarification on the difference in the Ethiopian health system between health posts and health centers.

It is interesting that the outcomes vary so appreciably by region and if the authors have valuable insight on why this then please include that in the discussion.

Lines 357-367 indicate that the analysis has implications regarding the need for youth friendly services, even though you found that age was not associated with FP counseling quality in the fully adjusted model. It seems that including parity and age in the same model washes out an association between age and the outcome, which is not surprising. I do not think the findings give a useful interpretation related to provider biases and clients’ age, and you should revise or remove this section from the discussion. Lines 419-430 are similarly confusing. Line 419 says age is not significantly associated with the outcome in the fully adjusted model, contrary to other studies and then you proceed to reflect on the crude OR in Model A. If after adjusting for parity, you find there is no age effect, then you might consider a different interpretation. For example, as women become more experienced with childbearing their knowledge and confidence during interactions with SRH providers increase and they are able to ask for and recall more information during their FP visits. I concede that women with little or no childbearing experience are also young, but I think you need to frame the discussion on age/parity and the outcome better and in a manner more aligned with Model B findings.

Limitations

Unless it is a formatting requirement of the journal, I think the limitations sections should be included in the discussion section (personally, I think it is best at the beginning of the discussion). I do not think it is a good way to end the article.

I think the final sentence of the limitations should be expanded to state, specifically, that the analysis lacks really critical supply-side variables, such as availability of methods at facilities, availability of trained staff, job aids, information leaflets for clients, etc. Also, the analysis doesn’t discuss whether survey participants were new or returning users, whether they had ever practiced FP in their lives, etc.

Conclusion and recommendation

The conclusions right now seem rather generic and disconnected from specific lessons from the analysis. This is understandable but the authors need to reflect on what they can credibly conclude and give guidance on based on their findings and report this in a more focused manner.

The study seems to conclude that having more formal education, access to FP information from media and more experience with SRH care seeking (based on higher parity levels) is associated with recalling more FP counseling information. However, we also know from the analysis that counseling tends to be better at health posts and health centers compared to hospitals and is particularly poor at pharmacies, that LARC provision is association with better counseling, that some regions do appreciably better than others and that overall, counseling quality has declined over time.

Though I am skeptical about this analysis because it does not include vital supply-side variable that affect counseling quality, I think the findings at least point in the direction of some supply- and demand-side actions that the MOH and partners should consider to address the problem of poor counseling quality.

A methodological recommendation to consider is incorporated more supply side measures (that PMA has made available from its extensive work conducting facilities assessments, I think) into future analysis on FP service quality.

7. PLOS authors have the option to publish the peer review history of their article (what does this mean?). If published, this will include your full peer review and any attached files.

Reviewer #1: **Yes: **Dr. Colin Baynes

---

## [Author Response · Author response to Decision Letter 1]

26 Mar 2022

Thank you for the opportunity to respond to reviewer comments. We thank the reviewer for a thorough secondary review. 

Additional Editor Comments:

This is an important analysis as I expressed in the previous iterations. The manuscript has improved and the authors addressed most of my comments/questions.

My previous question remaining or with unsatisfactory response:

1. I did ask about the missingness before. Clarifying, what is the response rate of these PMA surveys? What was done if there were participants with missing data among the variables included for this analysis? This could be placed between the current lines 77 to 83.

Thank you to the reviewer for catching this and we apologize for the oversight in the previous version. We have added the text below on page 3 in the revised clean manuscript.

- In PMA surveys considered in this analysis, the response rate is high across different survey years which ranges from 95.6% to 99.2%. The analysis was done for women who completed the survey. If a woman replied “Don’t know or No response”, these values were recoded as missing.

New few issues:

1. The reviewer below, points out the need for some introduction of the concept family planning (FP) service quality and how that end be measured as just 3 questions.

We have expanded our discussion of the Method Information Index in the manuscript. While we agree that three questions are unlikely to capture the full complexity of the concept of quality, the Method Information Index is a well-established indicator in the field of family planning and generally accepted as the current standard for measuring quality. We have included additional information on the creation and validation of the indicator in the Background section and in the abstract. 

“Counseling plays a key role in enhancing family planning services, providing contraception related information and supporting family planning and fertility goals for women of childbearing age [4]. The Method Information Index (MII) is the current standard proposed by the Track 20 initiative to monitor quality of family planning counseling [5, 6]. The simple index uses three questions -whether the respondent received counseling on multiple contraceptive methods, whether she was told about side-effects, and if so, whether she was told what to do about side-effects that are available in the majority of population-based surveys, including the Demographic and Health Survey (the DHS) and Performance Monitoring for Action (PMA) [7]. Adaptations have been proposed to the MII, including a fourth question assessing whether women were told they could switch to another method. This question was 

added in PMA 2019 Survey, however, and for trend purposes is not included in the current analysis.”

2. Line21: What is this coverage? Is this high quality of FP coverage?

 - Thank you. Yes, this has been updated.

3. Line 63: the SNNP abbreviation is used for the first time here. Can you write fully what it means?

 Yes done. SNNP stands for Southern Nations, Nationalities, and Peoples'

4. Just for clarification: was the year of survey included in the model as a categorical variable. Correct? Can you clarify that in table 1.

 - Yes, and updated accordingly.

5. Results related to table 2. Can you clarify how the weights of the two surveys were combined for 2014 analysis?

- Since the two surveys in 2014 conducted at the beginning and end of 2014, we recalculate the weights using the projected women in reproductive age (15-49) population in 2013.5 and 2014.5 which is 22326082 and 23198829, respectively. See the details below (under #7) how it is calculated. 

6. Lines 159: what were the conclusions of the Brant tests? Was the PO assumption kept across all variables?

 - Yes, we have included this on page 6.

7. The paragraph 171 to 174 states that all analysis used the sample weights. This leads to 2 questions:

a. How multiple survey weights were combined for this analysis?

- Following the recommendations by Ruilin Ren who is a senior sampling statistician at ICF, we recalculate the survey weights to compute the percentage of women who got high quality family planning counseling service as follows. 

denormalizedWeight=FQweight×(total females age 15-49 in the country at the time of the survey)/(number of women age 15-49 interviewed in the survey)

- Normalizeweight= denormalizedWeight/sum(denormalizedWeight)

- Where, FQweight represents the computed weight in each survey.

Year 2014 2015 2016 2017 2018 2019

Projected population of women 15-49 

23,198,829 

24,084,767 

24,949,201 

25,832,132 

26,725,476 27,620,498

b. It is very tricky to do random-effects models with survey weights. How the results would differ if you did not use weights. This could be placed in the supplements if indeed you did a random-effects model with sample weights.

- Thank you for the comment. As mentioned in line #140 and line #187 weights were used during the estimation of proportions and their respective confidence intervals. We did not consider sample weights when we fit the random-effects model.

8. The limitations should be part of the discussion as the reviewer points out below. Somewhere the last 3 paragraphs would be OK.

- Thank you. As most family planning research tends to include the limitations discussion near the end of the discussion section, we have inserted as the penultimate paragraph. 

- 

9. Thank you for the good plots. Just few things:

a. All of them should have the vertical y-axis start at zero. In ggplot2 there is something like expand_limits(y = 0) to sort that.

b. Make all of them to have the same maximum. It is easy to misunderstand the plots as they have different scales.

 - Thank you for the helpful comments. In the revised version, the figure axis was updated accordingly.

Reviewers' comments:

 Abstract

In methods section it is written “women aged 1549” but it should say “15 to 49”.

- Updated accordingly.

In results section what is meant by “the percentage of women that received high family planning counseling service…” Do you mean high quality FP services?

 - Yes, thank you. This has been updated throughout.

I am not sure if I use the term “determinants” to characterize the factors that you found were associated with FP counseling quality. For example, I think media is a contextual factor that may be associated with experiencing better counseling quality, but I question whether it truly determines the level of counseling quality.

 - We thank the reviewer for their suggestion, and have replaced “determinant factors” with “factors”. 

Even though this may cause the authors to exceed the word limit for the abstract, I think the abstract would be strengthened if it briefly stated how the authors define FP counseling quality.

- Thank you. We have added this to the introduction section of the abstract.

Background

The new paragraph that points out that in a study on MII across 25 countries “residence, education, household wealth and method type were generally found to be associated with receiving higher quality care” … To be helpful, please specify residence where and what method types were associated with higher quality care.

- Thank you. We have clarified in the introduction section that urban residence was associated with receipt of higher quality care and included additional information on method type. The updated language on page 2 is copied below. 

Despite substantial global efforts to improve conceptualization, measurement, and delivery of high-quality family planning services, studies consistently find that receipt of high-quality services is low. In a study exploring the levels and trends of the MII across 25 countries, urban residence, education, household wealth, and method type were generally found to be associated with receiving higher quality care [?]. In all countries combined, the median MII values show that the contraceptive information received by women varied by method type and was highest for implant users and lowest among women relying on sterilization.

In the same paragraph, when you say “individual level factors associated with receiving high quality care” it would be useful to clarify that you mean client-level (as opposed to individual provider-level).

- Thank you for this useful comment. We have rephrased the term “individual level” factors to “client-level” factors throughout the document.

Materials and Methods

Are you able to account for whether or not the survey participants were new or returning users? This matters since it is possible that returning users had already received the counseling information in recent FP visits at their nearby facility.

- Thank you for this question. The PMA surveys asked about the information that was received at the visit at which the client first received their particular contraceptive method. This has been clarified in the manuscript and the implications for the potential for recall bias has been included in the limitations section. We note that the PMA questions use the same question wording as the DHS when collecting information on the MII and thus, while this bias does exist, it is consistent with previous analyses. 

- 

To be helpful, specify what methods you define as long-acting and short-acting.

 Thank you. This has been clarified in the methods section. Specifically, we define long-acting methods as female sterilization, implant and IUD and short-acting methods as the injectable, pill, emergency contraception, male condom, the Standard Days Method, and lactational amehorrhea method (LAM).

Results

“More than 70% of women in all years got received their contraceptive methods…” Check for grammar.

- Thank you, this has been updated.

Line 258 on page 8: you mention a decline in the proportion of survey participants that report good counseling services, but you say it is a decline in coverage. Elsewhere you call it percentage of women received high quality FP counseling. I question whether coverage is the best term for this. I think it is a measure of the prevalence. Please be consistent in how you refer to the measure.

- Thank you, we have updated throughout.

Line 262 on page 8 references Table?? – please correct this.

 Updated.

Lines 255-267 page 9 – please check to see if you have used the correct tense.

Figure 2 label- “trends in getting high family planning counseling…” should it not say high quality family planning?

- Thank you. We have performed a thorough check of grammar throughout.

I understand the purpose of Table 4, but it is incredibly confusing. I suggest a table with a row for every year in the time series and columns for year, % change between successive years including the 95% CI in parentheses and cumulative % change between current and start year with 95% CI in parentheses.

- Thank you for the helpful suggestion. The table has been updated based on the following format.

Table X: Summary of observed percentage changes in high quality family planning counseling service across survey-years. 

Reference/ base year Survey year

 2015 2016 2017 2018 2019

2014 39.9(19.57,60.25) 16.6(-3.03, 36.2) 14.3(-8.52, 37.2) 5.1(-16.1, 26.3) -57.3(-67.11, -47.48)

2015 -16.7(-28.10, -5.27) -18.3(-34.7, -1.9) -24(-40.1, 9.7) -69.47(-76.5,-62.4)

2016 -2.0(22.02,18.17) -9.84(-28.44,8.7) -63.4(-71.9,-54.8)

2017 -8.1(-21.84,4.69) -62.65(-71.05,-54.25)

2018 -59.4(-68.56,-50.18)

Lines 278-286- there seems to be some repetition here… please check and revise accordingly.

- Thank you, this has been updated. 

Thanks for explaining the interpretation of Models A and B. After providing a thorough explanation of the interpretation for one of the explanatory variables, I do not think you need to repeat the same language over again for the other variables. But it is up to you.

-Thanks. This comment taken into account and we dropped the interpretation of some variables. 

Discussion

It would be interesting if the authors could elaborate further on what might be the drivers of the decline in the outcome variable over the span of the time series.

- Thank you. We have added the following statement.

“Except in Addis Ababa, in all other regions, high-quality family planning counseling peaked in 2015 and declined since then in all other regions. The observed relatively better family planning counseling service in 2015 may be due to the fact the Ethiopian health sector transformation plan started in this year [31]. As compared with other 

regions, high quality counseling service was low in Amhara region and sharply declined since 2017 (Figure 2). Nationally, our finding on the trend analysis in the percentage of women receiving high quality family planning counseling service declined overtime. This aligns with Hrusa's finding using PMA data [12].”

I think it is good that in the discussion you mention that the association you identified between media exposure and the outcome is likely due to the fact that the former enhances knowledge and may prompt clients to ask more questions. In link 382 of page 15 you say this supports women in receiving high quality services, but I think a better way of phrasing it is that media exposure enables a more informative client-provider interaction because it’s enhances client knowledge and empowers them to more inquisitive in the FP visit.

- Thank you, we have updated our language to better reflect the suggestion. 

Can you establish whether women’s reports of FP media exposure occurred before their reported FP visits in each survey? If not, then you should acknowledge that since women that received FP services may be more likely to seek out or recall FP-related media exposure (e.g., the visit peaked their interest, women that listen to educational media programs are more educated and therefore also more likely to use FP and ask questions during the visit). In other words, the direction of the association between media exposure and FP quality cannot be determined by this analysis, which limits the interpretation.

The abstract mentions that seeking FP counseling from pharmacies was associated with lower levels of FP counseling quality. This is an interesting finding and I think the authors should elaborate on this further in the discussion.

Thank you. In the discussion Section, we have added the following statement.

“Women who received care from pharmacies had significantly lower odds of receiveing high-quality family planning counseling than women who received care at either a health post or health center. In addition to differences in service modality and training of service providers, this likely reflects differences in characteristics of users and method choice; other studies have shown that women who use pharmacies tend to live in urban areas, be younger, and rely on short-term methods [38, 39].”

In the discussion, you mention that the odds of receiving high quality FP counseling are higher among women that obtain the service at public facilities compared to those that sought it at private facilities and refer to table 5 (lines 393-397 on page 16). But Table 5 does not include columns that refer to public vs. private facilities. Table 5 does give parameter estimates for health post, health center, pharmacy and other facility (ref hospital). This finding could be elaborated on in the discussion, with some clarification on the difference in the Ethiopian health system between health posts and health centers.

- Thank you. We clarify that the health posts, health centers, and hospitals included are all public health facilities, while pharmacies are all considered private facilities. We have provided additional information on page 12.

It is interesting that the outcomes vary so appreciably by region and if the authors have valuable insight on why this then please include that in the discussion.

-Added in the second paragraph of the discussion Section.

Lines 357-367 indicate that the analysis has implications regarding the need for youth friendly services, even though you found that age was not associated with FP counseling quality in the fully adjusted model. It seems that including parity and age in the same model washes out an association between age and the outcome, which is not surprising. I do not think the findings give a useful interpretation related to provider biases and clients’ age, and you should revise or remove this section from the discussion. Lines 419-430 are similarly confusing. Line 419 says age is not significantly associated with the outcome in the fully adjusted model, contrary to other studies and then you proceed to reflect on the crude OR in Model A. If after adjusting for parity, you find there is no age effect, then you might consider a different interpretation. For example, as women become more experienced with childbearing their knowledge and confidence during interactions with SRH providers increase and they are able to ask for and recall more information during their FP visits. I concede that women with little or no childbearing experience are also young, but I think you need to frame the discussion on age/parity and the outcome better and in a manner more aligned with Model B findings.

- Thanks for your fruitful comments. Accordingly updated.

Limitations

Unless it is a formatting requirement of the journal, I think the limitations sections should be included in the discussion section (personally, I think it is best at the beginning of the discussion). I do not think it is a good way to end the article.

- Thank you for the suggestion. We have included limitations in the discussion section, but have chosen to place them in the penultimate paragraph of the paper, as we found this was in keeping with the majority of family planning manuscripts included in our references. 

I think the final sentence of the limitations should be expanded to state, specifically, that the analysis lacks really critical supply-side variables, such as availability of methods at facilities, availability of trained staff, job aids, information leaflets for clients, etc. Also, the analysis doesn’t discuss whether survey participants were new or returning users, whether they had ever practiced FP in their lives, etc.

- Thank you. In the last line of the limitation paragraph, we included a statement which addresses the lack facility-level variables in the analysis and encourages further research on addressing this shortcoming. 

- 

Conclusion and recommendation

The conclusions right now seem rather generic and disconnected from specific lessons from the analysis. This is understandable but the authors need to reflect on what they can credibly conclude and give guidance on based on their findings and report this in a more focused manner.

The study seems to conclude that having more formal education, access to FP information from media and more experience with SRH care seeking (based on higher parity levels) is associated with recalling more FP counseling information. However, we also know from the analysis that counseling tends to be better at health posts and health centers compared to hospitals and is particularly poor at pharmacies, that LARC provision is association with better counseling, that some regions do appreciably better than others and that overall, counseling quality has declined over time.

- Thank you. We included more specific recommendations particularly highlighting the importance of providing additional training to pharmacists and related staff on contraceptive methods. 

Though I am skeptical about this analysis because it does not include vital supply-side variable that affect counseling quality, I think the findings at least point in the direction of some supply- and demand-side actions that the MOH and partners should consider to address the problem of poor counseling quality.

A methodological recommendation to consider is incorporated more supply side measures (that PMA has made available from its extensive work conducting facilities assessments, I think) into future analysis on FP service quality.

Thank you. We have added this recommendation in the limitations section.

---

## [Editor Report · Decision Letter 2]

20 Apr 2022

Trend and Determinants of Quality of Family Planning Counseling in Ethiopia: Evidence from repeated PMA cross-sectional surveys, (2014-2019)

PONE-D-21-19351R2

Dear Dr. Ejigu,

We’re pleased to inform you that your manuscript has been judged scientifically suitable for publication and will be formally accepted for publication once it meets all outstanding technical requirements.

Kind regards,

Orvalho Augusto, MD, MPH

Academic Editor

PLOS ONE

---

## [Editor Report · Acceptance letter]

19 May 2022

PONE-D-21-19351R2 

Trend and Determinants of Quality of Family Planning Counseling in Ethiopia: Evidence from repeated PMA cross-sectional surveys, (2014-2019) 

Dear Dr. Ejigu:

I'm pleased to inform you that your manuscript has been deemed suitable for publication in PLOS ONE. Congratulations! Your manuscript is now with our production department. 

Kind regards, 

on behalf of

Dr. Orvalho Augusto 

Academic Editor

PLOS ONE